# Effectiveness of Adherence to a Mediterranean Diet in the Management of Overweight Women: The Prospective Interventional Cohort Study

**DOI:** 10.3390/ijerph192315927

**Published:** 2022-11-29

**Authors:** Janka Poráčová, Ivan Uher, Hedviga Vašková, Tatiana Kimáková, Mária Konečná, Marta Mydlárová Blaščáková, Vincent Sedlák

**Affiliations:** 1Faculty of Humanities and Natural Sciences, University of Prešov, 080 01 Prešov, Slovakia; 2Institute of Physical Education and Sport, Pavol Jozef Šafárika University, 040 01 Košice, Slovakia; 3Department of Public Health and Hygiene, Pavol Jozef Šafárik University, 040 01 Košice, Slovakia

**Keywords:** Mediterranean diet, weight loss, determinants of health, healthy lifestyle, clinically significant weight loss

## Abstract

Evidence indicates that unhealthy eating habits constitute multilevel obstacles threatening health and well-being. Studies suggest that consumer choices have turned irremovably towards Western diets. The Mediterranean diet (MD) is considered one of the most effective in preventing and treating overweight and obesity, yet its results and associations are ambiguous. This explanatory research aims to examine the effect of the MD on anthropometric and biochemical variables in 181 females from an Eastern European country. The sample was divided into three distinct clusters based on age (tricenarian, quadragenarian, and quinquagenarian). Anthropometric and biochemical examinations in the three-month MD program plan failed to provide convincing evidence of the benefits of the MD on selected integrands. However, total body fat (FATP) values between groups showed a significant difference (*p* ≤ 0.032) between groups A and C (*p* ≤ 0.029), which can be attributed to the age of the cohort (30–39 vs. 50–60 years). Values in groups A and B (*p* ≤ 0.001) and C and A (*p* ≤ 0.001) were significant between the cohorts but did not indicate any changes in visceral fat (VFATL) in the individual groups. The presented findings can have implications for further investigation and the development of more comprehensive instruments, incorporating critical add-on constituents that will be appropriate to monitor, evaluate, and predict body weight management in experimentation.

## 1. Introduction

The current lifestyle of people in economically developed societies is characterized by a high availability of food and an increasing rate of physical inactivity. Unhealthy dietary patterns have been associated with the development of multiple age-related diseases [1,2]. Non-communicable diseases (NCDs) (mainly cardiovascular diseases, cancers, chronic respiratory diseases, and type 2 diabetes) are the leading causes of death worldwide. Their burden is expected to rise in the future. Indeed, treating and preventing these pathologies constitute crucial challenges for public health [3]. The NCD rise is due to a combination of genetic, physiological, and environmental factors and behavioral features, including unhealthy diets, physical inactivity, tobacco use, and excessive alcohol consumption [4]. Prediabetes and type 2 diabetes mellitus (T2DM) have become widespread globally in recent decades. It has now been widely demonstrated that the incidence of these pathologies is very high in overweight/obese patients. A study by Ivan et al. (2022) on a sample of 80 patients (40 men and 40 women) followed a very low-calorie Mediterranean diet (MD) and a very low-ketogenic MD for 30 days. Both diets (especially the ketogenic version) led to significant reductions in body weight and BMI and rapid improvements in metabolic, anthropometric, and body composition parameters in patients with prediabetes or diabetes and overweight/obesity [5]. In the last 50 years, other epidemiological and physio-pathological evidence has confirmed the beneficial effects of the MD on health and longevity. Several studies and meta-analyses of prospective cohort studies (more than 1.5 million subjects) showed that adherence to the MD correlated with improved health status, leading to a significant reduction in overall mortality (9%), mortality due to cardiovascular diseases (9%), tumor occurrence and progression (6%), and incidences of Parkinson’s and Alzheimer’s diseases (13%) [3]. A meta-analysis [6] reports that high adherence to the MD can have a positive impact on all parameters of metabolic syndrome.

In addition, there is sufficient evidence suggesting that the long-term consumption of an MD can protect from obesity and improve cardiometabolic risk markers. The meta-analysis included seven prospective cohort studies which revealed that greater adherence to an MD was significantly associated with a 9% reduction in the risk of being overweight or obese. Moreover, a significant inverse association was found between the MD score and 5-year weight gain. Considering the pandemic prevalence of overweight and obesity, even a minimal advantage of MD adherence to reduce this prevalence would have a substantial clinical impact on the entire population [7]. Because of the significant public health challenges of overweight and obesity, the serving sizes of the MD should be frugal and moderate to adapt to modern, increasingly sedentary lifestyles [8]. Accordingly, mounting evidence has demonstrated that moderate calorie restriction, without malnutrition, exerts beneficial effects [9]. Some investigations [3,10,11,12] assert that the MD hinders overweight and obesity-related diseases [13,14]. Esposito et al. (2011) [15] in their study on the MD and weight loss (a meta-analysis with 3436 participants) indicated the significant effect of the MD on body weight (mean difference between MD and control diet: −1,75 kg, 95% CI; −2.86 to −0.64 kg) and body mass index ((BMI) mean difference: −0.57 kg/m^−2^, −0.93 to −0.21 kg/m^−2^). The effect of the MD on body weight was larger in association with energy restriction (mean difference, −3.88 kg, −6.54 to 1.21 kg), and increased physical activity (−4.01 kg, −5.79 to −2.23 kg). Out of 40 diets, the MD has been voted as the best diet for the fourth year in a row [16]. Several studies over recent decades have shown that the MD is significantly associated with diverse health outcomes such as good health status, better biochemical profile, and quality of life [17]. However, the epidemiological evidence supporting a causal link between the MD and body weight is inconclusive. Moreover, the implementation of the MD in culturally diverse environments can be dissimilar. The MD is a food philosophy inspired by the eating habits of the population who live in countries bordering the Mediterranean Sea; these differ slightly, resulting in modified versions of the MD. The MD emphasizes eating fruits, root vegetables, grains, legumes, nuts, seeds, and olive oil, which provides a high content of monounsaturated fatty acids and a reduced intake of saturated fatty acids [18]. Furthermore, the MD characterizes a moderate intake of red wine during meals, an average intake of seafood, poultry, and dairy products, and a low consumption of red meat and sweets. The MD does not emphasize specific amounts of foods or portion sizes. A combination of MD foods appears protective against disease, as the benefits are not as substantial when looking at single foods or nutrients included in the MD [19]. Two additional aspects to be included with the MD include daily physical activity (in activities of daily living) and adequate sleep and hydration [20]. We expected that after the three-month intervention (MD) we would observe a reduction in body weight and a favorable outcome of anthropometric and biochemical indicators in the selected cohort sample.

## 2. Materials and Methods

### 2.1. Study Participants

A total of 181 overweight females residing in Slovakia, with ages ranging from 30 to 60 years old, participated in this study over three months (May–July 2022). The participants were recruited through a newspaper advertisement in a regional paper about the possibility of participation in a weight loss program over a three-month period. Via a questionnaire, we obtained information about their health conditions. From 215 applicants, 34 subjects were excluded following cardiometabolic abnormality. The representative sample was branched into three distinct clusters. Cohort A consisted of 59 females aged 30–39 years old, B consisted of 62 females aged 40–49 years old, and C consisted of 60 females aged 50–60 years old. Visually display sequence of activities depicts Figure 1.

Participants did not receive any kind of compensation or benefit for their participation. The test sample (n = 181) was calculated by sample size calculation. Sample size = (Z-score)^2^ × SD × (1 − SD)/(margin of error)^2^. Sample size = 3.84 × 12.41 × 11.41/52.56. The sample size was 10.34. Alongside sample size calculation, we determined the sample size by imitating a sample size of similar studies.

### 2.2. Ethics

Ethical approval for the study was obtained from the University Ethics Committee (UEC) of the University of Prešov in Prešov, Slovakia. No: ECUPO42022PO-1/7.

### 2.3. Biochemical Examination

Comprehensive biochemical examination at the baseline incorporated LDL markers (low-density lipoproteins), HDL (high-density lipoproteins), CHOL (total cholesterol), TAG (triacylglycerides), GLU (glucose), non-HDL, urea, uric acid, ALT (alanine aminotransferase), AST (aspartate aminotransferase), GGT (gamma-glutamyltransferase), and CRP (c-reactive protein). Blood collection: a venous blood sample was taken voluntarily in the morning on an empty stomach in the presence of a medical professional at the University of Prešov. Biochemical analysis was executed using a fully automated biochemical analyzer: Cobas Integra 400 plus. The biochemical analysis of the blood dichotomized the subjects to be accepted or declined to participate in the study (i.e., with or without metabolic or cardiovascular risk).

### 2.4. Anthropometric Analysis

Anthropometric analyses were performed using bioelectrical impedance (Quadscan 4000 Touch Screen) BodystatQuadScan 4000 Touch Screen for measuring body weight, percentage of total body fat (FATP), visceral fat (VFATL), muscle mass (FFM), and total body water (TBW). Other monitored parameters included body mass index (BMI) and basal metabolic rate (BMR). Participants were measured in the morning at the beginning of each month during the three months of the experiment.

### 2.5. Questionnaire of Preferences

A personalized questionnaire was conducted on the research participants at the first consultation by a dietician. The questionnaire consisted of 3 distinct parts. In the first, the proband filled in personal data. In the second, the researcher measured the subjects’ basic anthropometric parameters (body weight, body height, waist, chest, hip, neck, thigh, and arm circumference, blood pressure, and blood pulse). Consequently, they were given an identification code under which the actual measurement was performed by the bioimpedance method using the BodyStatQuadScan 4000 TouchSreen instrument. The third part of the questionnaire focused on family history, previous and current diseases, and treatment. Additionally, questions also concerned the participants’ dietary preferences and frequency of physical activity.

### 2.6. Personalized Diet Plan

The participants adhered to a three-month personalized diet plan executed to achieve weight loss. The MD was individualized to every subject based on the biochemical blood analysis anthropometric assessment and the questionnaire where they indicated an individual dietary preference. Participants with higher cholesterol values were advised to lower their fat intake, stress amount, and food portion size. Subjects with higher blood glucose levels had a diet emphasizing a balanced intake (food volume and portion size) of carbohydrate/glucose units throughout the day, suggested by the dietitian, with the consequent effect of lowering glycemia. For participants with elevated liver test values, a sparing diet (avoiding uncooked shellfish, saturated fat, refined carbohydrates, and salt) was indicated. To all individuals, the purpose of the investigation was explained. With a signature, the participants confirmed their agreement with the terms and conditions of the examination. Participants that had previously had a metabolic or cardiovascular disease or were currently suffering from a metabolic or cardiovascular disease were excluded from the research. An investigation was conducted anonymously with unrecorded personal data. The probands attended regular monthly controls, which included the measurement of anthropometric parameters and consultation with a nutritionist.

### 2.7. Data Analyses

All data were analyzed using SPSS (version 20.0, SPSS: Armonk, NY, USA) and the program RStudio. The results are presented as mean ± SD. Changes in body weight, FFM, and TBW were calculated and expressed in kg, BMI in kg·m^2^, FATP in percentage, VFATL in scale level, and BMR in kcal. The values of the variables were normally distributed. Descriptive statistics were broken down into measures of central tendency and variability (spread). Paired t-tests were used to test for differences between baseline, each month of the three months, and the end of the three months. Differences were considered significant at *p* < 0.05.

## 3. Results

### 3.1. Participants’ Characteristics

There were no differences in ethnicity, background, culture, or dwelling at the inclusion. No significant disparities in education were observed, which altogether creates the assumption of a more homogeneous representation. The participants’ main objective was voluntary weight loss associated with the MD, concurrent with additional health benefits.

### 3.2. Biochemical Indicators

Screenings of urea, creatinine, uric acid, ALT, AST, and GGT revealed optimal kidney and liver functioning, as well as normative triglycerides, HDL, non-HDL, and LDL levels in most of the participants. There were no significant differences between the biochemical characteristics of the subjects at the baseline, excluding total CHOL linking group C 5.4 mmol/L, that had a borderline level. The blood glucose level was in the normal range. Plasma C-reactive protein did not hit the inflammation risk thresholds. From the above, we can extrapolate the adequate health and homogeneity of the sample that can support a determination of causality (cause and effect) (Table 1).

### 3.3. Anthropometric Characteristics

All subjects (all three cohort groups) were overweight during the entire investigation. Body weight reduction was observed progressively in groups B and C (−3.4 kg (−4%) and −2.1 kg (−3%)); body weight reduction was not detected in group A after the three-month follow up. BMI in group A was 26.7 kg·m^−2^ with no improvement detected, and 27.9 kg·m^−2^ in group B with 5% improvement. Group C had a BMI of 27.8 kg·m^−2^, enhanced by 2.5% in congruence with the weight losses of the sample. All subjects (all three cohort groups) were overweight during the entire investigation. FATP did not change convincingly throughout the experimentation in all cohorts. However, we observed significant differences between groups A and C (*p* ≤ 0.032 vs. *p* ≤ 0.029).

VFATL should be below level 13, and the recommendation suggests maintaining a visceral fat lower than ten or a visceral fat area under 100 square centimeters. The present examination detected A at 4,5, B at 7, and C at 9 at the baseline, and the value did not change throughout the investigation (Table 2). Accordingly, the modification of VFATL was not determined. This can be attributed to the insulin resistance (GLU) in group A (5 mmol/L vs. group C 5.4 mmol/L) (borderline) presented in Table 3. Moreover, regular aerobic exercise can decrease visceral fat, and enhanced sleep and reduced stress can potentially contribute to the possible outcome. A healthy percentage of FFM in women is 68–80 percent. FFM remains relatively stable until approximately age 50, with a slower rate of decline (about 16% between the ages 25 and 70 at a rate of −0.16 kg/year). If a person is obese they have a lower percentage of FFM and a higher percentage of FM, translating to a decrease in TBW.

There is a small but not significant negative linear association of TBW to weight decline with age in females. The mean ratio of TBW to weight declines with age as a function of increased body fat. The average healthy range of TBW for women is 40–45%. Our observation points to the middle range of 35.7–37.7%, which was below average across the entire examination, and corresponds with an accumulation of adipose tissue in our cohort samples. An average female has a BMR of around 1410 kcal or 5900 kJ. The BMR of our selection ranged from 1538 to 1497 (kcal). Weight loss, tissue loss, and metabolic adaptations reduce resting metabolic rate (RMR). BMR is usually slightly lower than RMR measurement. A more accurate Mifflin-St Jeor equation of RMR showed that the average BMR of the entire sample was 1323 kcal, which is about 200 kcal lower than a calculation of BMR. Moreover, the average metabolic age (Harris–Benedict formula) of the entire sample was about five years higher than the actual age throughout the investigation (46 vs. 51), which indicates a need for lifestyle modification.

FFM, TBW, and BM values decreased during a three-month program in each group (Table 2). BMI values slightly decreased in groups B and C, but there was no change in group A. Body fat decreased in cohorts B and C, but increased in cohort A, and visceral fat values remained unchanged.

The optimal proportion of total body fat in young females is 18–25%. In older females, physiologically, it can reach 30% of the whole body weight. From the measured values before and after the 3-month program, it can be stated that our sample had an above-average body fat percentage. Visceral fat is closely linked to overweight, obesity, hypertension, and hyperlipidemia, leading to serious chronic diseases, particularly diabetes type 2, and acute conditions such as myocardial infarction. The ideal value of visceral fat ranges from 1 to 3. In our analyzed groups, visceral fat values were not lower than 4. While adhering to the lifestyle of the MD for weight reduction, the resulting bioelectric impedance measurement showed that cohorts B and C reduced their body weight by −3.4 kg and −2.1 kg, respectively. No changes were observed in group A. Overweight was observed continuously through the entire duration of intervention in all three cohort groups. As mentioned above, substantial differences in BMI measurements were not noticed.

Table 3 shows the correlation analysis of the individual significant parameters for the overall data (N = 181) at the baseline of the three-month intervention. All parameters between groups were statistically significant. The largest statistically significant differences were between groups A and C, but this can be explained by the largest age difference. The VFATL parameter was significant between groups A and B, and C and A; the CHOL parameter was significant between groups C and A, and C and B.

Correlation analyses of individual parameters for the total data (N = 181) at baseline and at the end of the three-month intervention are presented in Table 4. After the three-month intervention, we can see statistically significant differences for the TBW and VFATL parameters. The BMR parameter with a *p* value of 0.07 perhaps would have been statistically significant with a larger observation set.

## 4. Discussion

Based on these findings, it can be argued that additional factors contributed to our cohort sample’s body weight reduction. Healthy weight loss and its maintenance may not be entirely based on nutritional therapy. There are additional integrands (Table 5) that need to be considered that may explain the process, strength, and direction of this relationship.

Crous-Bous et al. (2014) [19] postulate that that the MD is more of a guideline than a diet in contemporary society (it does not exclude any food groups), but is more akin to plant-based eating [4]. Although the pyramid shape advocates the proportion of foods to eat, it does not define portion sizes or determine amounts. It is up to the individual to choose how much food to eat at each meal. Wattles (2012) [27] distinguishes between “hunger and appetite”, and postulates that everything we eat after we have satisfied our hunger, we eat only to satisfy our taste and appetite. Furthermore, he claims that the habit of eating for sensual pleasure is very deeply rooted in human beings. The author concludes that we should eat when we experience a “deserved hunger,” a logical indication that it is the proper time to eat, and that any other time is unmerited. To make a change, individuals must implement this perspective into practice. Subjects were instructed on what and how much to eat (to satisfy hunger).

Nevertheless, it was up to the participants how they interpreted physical hunger or emotional cravings, which could ultimately influence the outcome. Robinson et al. (2021) [25] claim that deficits in interoception (processes by which we sense, interpret, and integrate signals originating from within the body) are associated with less reliance on satiety signals. A growing body of research suggests that the interoceptive process (relating to emotional regulation) may play an essential role in shaping human health. Deficits in interoception have been linked to higher BMI and may contribute to weight gain [26].

However, it remains unclear whether deficits or sensibilities to interoception contributed to or were a consequence of the weight gain of the sample. Müller et al. (2016) [28] provide evidence that a higher BMI of 5 kg·m^−2^ is associated with an increase in overall mortality. Even though being overweight is a risk factor for specific medical conditions, including coronary heart disease type 2, diabetes, and some forms of carcinomas, the relative risk of mortality from these conditions among the overweight is not necessarily as high as the relative risk of incidence. Flegal et al. (2013) [29] claim that survival for people with certain diseases may be slightly better for heavier subjects. Still, the underlying reasons for increased survival among overweight people have not been well elucidated. Potential indications incorporate both physiological and behavioral influences. According to Myrskylä and Chang (2009) [30], small weight gains or losses seem insignificant among normative and overweight individuals, and are potentially harmful only among those well into the obese range. A modest weight loss of 5–10 percent of the total body weight over six months is likely to produce health benefits, such as the normalization of blood glucose and cholesterol concentration [31]. The standardized bioelectrical impedance (SPhA), which is related to mortality [32], showed that the average value of BMI of 27 kg·m^−2^ in the present sample may lead to modest survival benefits of being overweight in a variety of adverse situations (chronic, acute illness, and medical procedures). However, this may not agree with self-related health (SRH) and general life satisfaction (LS), which are more strongly correlated to perceived weight status and the motivation to join our program.

Irrespectively, visceral fat (active fat) is strongly linked to metabolic disease. Subcutaneous fat does not carry those same risks (some may even be protective). A study by Mittal (2019) [33] showed that losing 6–7% of total body weight can reduce both subcutaneous and visceral fat. According to the Centers for Disease Control and Prevention [34], losing around 0,5 kilos of body weight in a week is ideal, which results in 2 kilos in a month. Losing body weight can be surprisingly simple and unusually challenging at the same time. The decrease in body weight in groups B and C (4%-3%) (not detected in group A) after the completion of the program was the result of intentional behavior, i.e., the MD, presumably due to the loss of FFM, TBW, and a decrease in BMR. The MD is considered a more balanced eating plan than the ketogenic diet. Some examinations indicate that TBW volume, on average, maintains a reasonable degree of stability in women through a large portion of adulthood [35]. Investigations [36,37] have shown that a low-carb diet is more effective at reducing visceral fat than a low-fat diet. Furthermore, contrary to popular belief, it is not skeletal muscle but rather adipose tissue losses that seem to drive RMR reduction following weight loss [38]. Furthermore, Yener et al. (2017) [39] assert that cortisol can increase the visceral fat the body stores so that stress reduction may influence fat loss. VFATL in the sample was rated between 1 and 12, indicating a healthy (above average) visceral fat level.

The sample included women with higher education and economic security. Higher education and socioeconomic status can be associated with better functional health literacy [26]. This may have favored participation, but did not guarantee the outcome of our intervention. Furthermore, poor sleep quality is linked to a higher risk of various chronic conditions, from heart disease to depression [20]. Kroese et al. (2014) [40] claim that bedtime procrastination may also lead to unhealthy behavior as an essential factor related to having insufficient sleep, consequently affecting individual health and well-being, and thus could appear as a potential contributing factor to the result of the investigation. Lastly, physical activity and exercise participation are associated with healthy outcomes [23]. Undoubtedly, physical activity and a regulated exercise regimen can reduce alimentary obesity by increasing total energy expenditure, thus promoting a negative energy balance when caloric intake is lower than energy expenditure. Weiss et al. (2016) [41] assert that minimum levels of physical activity (150 min of moderate-intensity exercise training) without dietary restriction may induce modest weight loss (≥5% weight loss). However, Johns (2014) [32] argues that there is no significant difference in weight loss from 3 to 6 months between diet-only and exercise-only programs, although a more pronounced weight loss has been detected after the completion of 12-month combined programs. In sum, the optimal strategy for promoting weight loss is combining a behavioral weight management program and adherence to adequate (continuous) amounts of physical activity or exercise regimens. Considering the 3-month program based on the above findings, we may conclude that excluding a physical training program did not affect our research conclusions. Furthermore, some investigations claim that a low-carb diet is more effective for maintaining health than a low-fat diet [14,29].

Furthermore, research shows that creating and sustaining a healthy lifestyle is challenging [24]. Healthy weight loss is not entirely about diet (eating less). Instead, one can realize that healthy body weight is a consequence of compelling lifestyle modification strategies, specifically concerning the pillars (eating a healthy diet, getting enough sleep, being physically active, having meaningful relationships) that affect our thoughts and emotions [42]. The presented emphasis is on a holistic view grounded in bio-psycho-social and environmental determinants. To this effect, healthy, sustainable eating should prevail over a particular diet plan.

The presented study has potential limitations. The evaluation of a diet using any method will link to measurement errors. This error is related to being random and would tend to underevaluate the connection between the intake of products and body weight modification. Furthermore, compliance with the MD regime can be challenging. We cannot rule out the possibility of residual confounding factors contributable to personal, social, and environmental influences. The study population comprised educated individuals that were motivated to lose body weight. Even though it is improbable that the biological processes underlying this association are dissimilar in other people, our results may not be generalizable for a broader group of people or situations. 

To establish complex conceptual and statistical models that include moderators and mediators to improve objective and perceived measures of the built environment and strengthen the evidence of causality, a more comprehensive research design is needed. A strength of our study is that it incorporates repeated measurements of anthropometric and biochemical characteristics. The monthly structured observation provides a view of reliable markers. Coherent results across the three independent groups were delineated by the range of the females’ ages. Future research should target thoroughly personalized strategies addressing the predominant causes of overweight and its treatment possibilities.

## 5. Conclusions

The main objective of this prospective interventional study was to examine the effect of the MD on body weight in a distinct female sample of an Eastern European country. 

During the three-month intervention MD program, the investigation did not observe the expected loss of body weight, visceral fat, or body fat. The presented results indicate that additional characteristics or circumstances may have contributed to the outcome (body weight reduction). Healthy weight loss and its maintenance are not entirely based on nutritional therapy. Additional integrands need to be considered that could potentially contribute to elucidating the processes, strengths, and directions of the underling determinants. This paper provides a platform for further research to contribute to a better understanding of body weight loss management and its underlying determinants through prospectively planned cross-sectional, longitudinal studies. It offers an innovative paradigm before monitoring and evaluating an MD or congruent dietary strategy. Due to the limitations of the data, anthropometric characteristics, biochemical indicators, and personalized MD plan, future studies should take into consideration other constituents (interoception, sleeping patterns, emotional health, environmental integrants, and more) to further elucidate the complex interplay between diet regiments and their efficacy in confronting weight management. Conceivably, a combination of physiological, psychological, and socioeconomic factors (holistic view) contribute (initiate, modify and sustain) to habitual changes.

## Figures and Tables

**Figure 1 ijerph-19-15927-f001:**
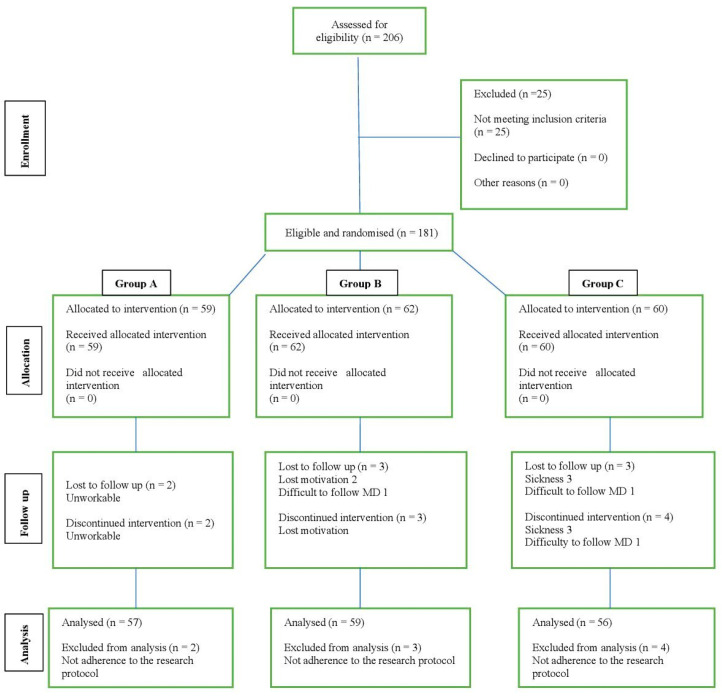
Participant flow diagram (www.consort.statment.org, accessed on 2 November 2022).

**Table 1 ijerph-19-15927-t001:** Biochemical analysis of the blood samples.

Parameters	Total (N = 181)	Group A (N = 59)	Group B (N = 62)	Group C (N = 60)
UREA (mmol/L)	4.4	4.1	4.3	5.3
CREATININ (mmol/L)	68.3	67.5	69.0	67.5
URIC ACID (µmol/L)	280.5	262.3	275.0	290.5
ALT (µkat/L)	0.33	0.31	0.33	0.40
AST (µkat/L)	0.35	0.35	0.32	0.36
GGT (µkat/L)	0.30	0.26	0.34	0.35
CHOL (mmol/L)	4.9	4.7	4.8	5.4
HDL (mmol/L)	1.60	1.65	1.48	1.65
Non-HDL (mmol/L)	3.29	2.96	3.26	3.73
LDL (mmol/L)	2.98	2.67	2.93	3.23
TAG (mmol/L)	1.09	0.99	1.18	1.11
GLU (mmol/L)	5.1	5.0	5.1	5.4
CRP (mg/l)	2.0	1.6	2.3	2.2

Abbreviations: ALT—alanine aminotransferase; AST—aspartate aminotransferase; GGT—gamma-glutamyltransferase; CHOL—total cholesterol; non-HDL—total cholesterol minus HDL cholesterol level; LDL—low-density lipoproteins; TAG—triacylglycerides; GLU—glucose; CRP—c-reactive protein; data are expressed as mean (SD) unless otherwise indicated.

**Table 2 ijerph-19-15927-t002:** Results of bioelectrical impedance during three months of experimentation for all three groups.

Parameters	Total (N = 181)			Group A (N = 59)			Group B (N = 62)			Group C (N = 60)		
**Measurements**	**1.**	**2.**	**3.**	**1.**	**2.**	**3.**	**1.**	**2.**	**3.**	**1.**	**2.**	**3.**
**AGE (years)**	46	46	46	30.5	30.5	30.5	45.5	45.5	45.5	57	57	57
**WEIGHT (kg)**	79.1	77.1	76.6	73.65	73.55	74.2	81.35	79.85	77.95	78.9	77	76.8
**BMI (kg·m^−2^)**	27.8	27.4	27	26.9	26.65	26.6	28.45	28.25	27	28.3	27.5	27.6
**FATP (%)**	36.1	35.9	35.4	32.75	33.5	33.85	36.45	35.9	34.9	38.5	37.5	38.2
**VFATL (level)**	7	7	7	4.5	4	4.5	7	7	7	9	9	9
**FFM (%)**	50.7	50.2	49.5	50.7	50.45	50.15	51.6	50.7	49.75	49.9	49.4	48.8
**TBW (%)**	37.1	36.8	36.2	37.1	36.9	36.75	37.75	37.15	36.45	36.5	36.2	35.7
**BMR (kcal)**	1538	1523	1497	1543	1534	1516	1568	1530	1503	1502	1492	1471

Abbreviations: BMI—body mass index; FATP—body fat; VFATL—visceral fat; FFM—muscle fat; TBW—total body water; BMR—basal metabolism rate.

**Table 3 ijerph-19-15927-t003:** Correlation analysis of individual significant parameters for the overall data (N = 181) at the baseline of three-month intervention.

Parameters	*p*-Value	Difference between the Groups
**FATP**	0.032	C—A (0.0291776)
**VFATL**	1.49 × 10^−9^	A—B (0.0000274) C—A (0.000000)
**CHOL**	7.59 × 10^−5^	C—A (0.0001434) C—B (0.0015425)
**non-HDL**	0.000497	C—A (0.003301)
**LDL**	0.00412	C—A (0.0029018)
**GLU**	0.000257	C—A (0.0001445)

Abbreviations: *p* < 0.05 level of significance; FATP—body fat; VFATL—visceral fat; CHOL—total cholesterol; non-HDL—total cholesterol minus HDL cholesterol level; LDL—low-density lipoproteins; GLU—glucose.

**Table 4 ijerph-19-15927-t004:** Correlation analysis of individual parameters for the overall data (N = 181) at baseline and the end of the three-month intervention.

Parameters	*p* Value
BMI 1–BMI 3	0.2744
TBW 1–TBW 3	0.04238 *
BMR 1–BMR 3	0.07668
FATP 1–FATP 3	0.9018
VFATL 1–VFATL 3	0.0001416 *
WEIHGHT 1–WEIGHT 3	0.2718

Abbreviations: * *p* < 0.05 level of significance. BMI—body mass index; FATP—body fat; VFATL—visceral fat; TBW—total body water; BMR—basal metabolism rate.

**Table 5 ijerph-19-15927-t005:** Factors contributing to body weight management.

Combination of foods	Crous-Bou M et al. 2014 [21]
Sleep pattern	Eugene A, Masiak J 2015 [20]
Bedtime procrastination	Kroese FM et al. 2014 [22]
Physical environment	Ding D, Gebel K 2011 [23]
Physical activity	Swift DL et al. 2018 [24]
Stress management	Ornish D. 2021 [25]
Interoception	Robinson E et al. 2021 [26]
Higher education	Vranes AJ, Mikanovic VB 2012 [27]

## Data Availability

The authors have full access to all specific material used in this paper and take responsibility for the use and accuracy of the information provided.

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
