# Peer review of "Effectiveness of Adherence to a Mediterranean Diet in the Management of Overweight Women: The Prospective Interventional Cohort Study"

_ijerph, 2022, doi:10.3390/ijerph192315927_

Round 1

Reviewer 1 Report (New Reviewer)

The manuscript entitled: „Effectiveness of Adherence to a Mediterranean Diet in the management of overweight women: the prospective interventional cohort study” presents scientific and practical importance in the medical and nutrition fields. The Mediterranean diet has long been considered the most effective in preventing and treating overweight and obesity. The benefits of MED for weight loss and a reduction in the incidence of cardiovascular events have been identified in several studies.

The article needs improvement. I have attached several observations that I hope will be useful in improving the manuscript: 

- Materials and Methods section requires restructuring. The trial registration number should be provided. A sequence of methodological stages (steps) and a figure summarising the process should be indicated (flow diagram of subjects’ participation in the trial).

- Sample size calculation: authors should provide a detailed description of the sample size calculation in the section of the study population.

- Adhering to the prescribed diet is essential to losing weight and improving your metabolic health. How was diet adherence checked?

- Tables 2 and 3: no reference in the description of the results.

- Line 147-152 and 152-157: the sentences are repeated. Rewrite the sentences with more clarity.

-  In the discussion, the publication number in the reference list was not included next to the author's names.

- References should be checked for uniformity in the formatting style and journal names.

- Editorial correction needed.

- Conclusions: the problem discussed is scientifically relevant, although some of the conclusions reached seem to be expected.

Author Response

Thank you for your valuable and stimulating comments on our submission, which will help us broaden our perspective in the context of the paper and beyond. 

Thank you once more.

Authors of the manuscript

Reviewer 2 Report (New Reviewer)

Dear authors,

I reviewed your article - Effectiveness of Adherence to a Mediterranean Diet in the management of overweight women: the prospective interventional cohort study - which aim was to examine the effect of MD on anthropometric and biochemical variables at females from Eastern Europe country.

I have some suggestions for your article:

1. Mention in Introduction more studies concerning MD at persons, especially womens from Europe and Eastern Europe.

2. At the end of introduction mention the aim of the study.

3. References must be numbered in order of appearance in the text and listed individually at the end of the manuscript.

4. Table 4 and text form lines 160-170 must be included at 3.2 where you present Biochemical Indicators.

5. Tables 2 and 3 are not mentioned in text.

6. You have 4 consecutive tables without text, I recommend to separate them and to introduce some specific text for each.

7. References 3, 8, 25, 35 at too old, use more recent related articles (from last 5 years).

Author Response

Thank you for your valuable and stimulating comments on our contribution, which will help us broaden our perspective in the context of the paper and beyond. 

Thank you once more.

Authors of the manuscript

Round 2

Reviewer 1 Report (New Reviewer)

The authors made the changes. The article can be accepted.

Best regards.

This manuscript is a resubmission of an earlier submission. The following is a list of the peer review reports and author responses from that submission.

Round 1

Author Response

Thank you for the valuable comments. We appreciate that. 

  1. Review

A first remark is that I miss the innovative aspect of the study a little bit. Maybe this can be added in the introduction. Why was this study performed while there is already a lot of literature about the MD? The introduction is also missing some ‘body’. Please add some concrete results/numbers about the association of the MD and health outcomes. End the introduction with the problem statement followed by the purpose of the present study. Modified Line 16, 23-28

Line 13-14: … threatening our health and well-being … Corrected Line 13-14

Line 20-21: Anthropometric (weight, BMI, FATP, VFATL, FFM, TBW, and BMR), biochemical examinations (urea, creatinine, uric acid, ALT, AST, GGT, CHOL, HDL-CH, non-HDL, LDL-CH, TAG, GLU, and CRP) what are the abbreviations stand for?

Abbreviations are given in the text. Lines 80-84

Line 23, 45, 146, 155, 182, 190, 217, 223, 239, 269, 288, 294, 301, 311: avoid the use of the word ‘we’. Please reformulate the sentence. Revised throughout the paper

Line 23-25: This sentence is not clear to me. Please explain it better. Sentence was modified Lines 23-26

Materials and methods: how were the participants recruited? By mail, social media, via which canal? How was representativity measured?  Lines 68-75

Line 73: Antropomethric analyses Wwere performed using bioelectrical       impedance …             Corrected Line 94

Line 77: … were measured … morning… A word is missing here. Each morning? Corrected Line 98

Line 80: … was implemented on the participants … is this a correct sentence? Corrected Line 101

Line 82, 83, 88, 93, 94, 95 226: is ‘proband’ or ‘partaker’ a good term? Maybe ‘participant’ is better? Modified throughout the whole paper

Line 84-86: They were given Consequently gave them an identification code consequently under which the actual measurement was performed by the bioimpedance method using the BodyStat QuadScan 4000 TouchSreen instrument. Corrected Line 105

Line 92: The MD dietCorrected Line 113

Line 94-95: They were given a low-fat diet in general? So also no unsaturated fats like in the MD? Modified Lines 116-117

Line 95-98: Item previous remark. Is the Mediterranean diet not suitable for all these medical   conditions, without individual adaptations? MD is suitable for all medical conditions. We assumed that individual adaptation enhance adherence to MD.

Line 99: What is sparing diet. Sparing diet means recommended diet for subjects with marginally higher liver enzymes and proteins (avoiding uncooked shellfish, saturated fat, refined carbohydrates, salt) was recommended. Line 118

Line 99-100: With signature and confirmed their agreement with the terms and conditions of the examination.  This doesn’t look like a correct sentence.  Modified Line 120

Line 100-101 Participants that suffering from this metabolic or cardiovascular disease were excluded from the research. Does a high cholesterol value belong to a cardiovascular disease? Because the participants with high cholesterol values were included.

Lines 121-124.  Cholesterol in the groups A and B were normative. Group C exhibited borderline value. (Overall, we accepted that as an admissible value).

Line 104: The Rresults are presented … Corrected Line 127

Line 105: Changes in body weight … Corrected Line127

Line 106: … and BMR in (kcal). Corrected Line 129

Line 115-118: The participants’ main objective was voluntary decrease in body weight (diet plan) the  incidence of civilization diseases (obesity, diabetes mellitus, cardiovascular diseases), emeliorate health and well-being.  There seems to be something wrong with the sentence. Modified Lines 138-139

Line 120: Screening of (urea, creatinine, uric acid, ALT, AST, and GGT) revealed. Corrected Line 141

Line 122-124: There were no significant differences between biochemical characteristics of the subjects at the baseline, excluding total CHOL linking group C 5,4 mmol/l borderline level A 4.7mmol/l B 4.8 mmol/l desirable. There seems to be something wrong with the sentence. Modified Lines 142-144

Line 128-191: Please just add results from the present study in the results section without interpreting or discussing thing. The interpretation and discussion of the results belongs to the discussion section. So remove all the references in the results section and use them in the discussion section. Please add p-values where necessary in the text of the results section. Changed Lines 255-261, 276-281

 Line 136-142: Was this statistically tested? Because when there is no statistically significant difference, there cannot be spoken about a difference. Yes.

Line 144-145: Our The present examination detected A 4,3 - B 7 and C 9 at the baseline and the values did not change throughout the investigation (Table 1.) Corrected Line

Line 144-145: I suppose that this is also statistically tested? So please add p-values.

The FATP parameter is reported in Table 2. FATP is significant at p ≤ 0.032. Between groups A and C p ≤ 0.029. Line 159.

 Line 147: … to insulin resistance GLU in group A (5 mmol/l vs. group C 5,4 mmol/l) (borderline) presented in (Table 4.). Corrected Lines 161-162

Line 154-155: … age 50 with a slower rate of decline (about 16% between the ages 25 and 70 at a rate of -0,16kg/year). Corrected Line169

156: If it wasn’t statistically significant, you can’t speak about a drop. Corrected Line 171

Line 167-168: How did you get to this one value? This is not so clear. Please explain it better.

BMR results were determined by the BodyStat QuadScan 4000 Touch Screen analyzer. Study participants (N=181) had an average basal metabolic rate of 1538 kcal at baseline examination. At the end of the study, participants had values of 1497 kcal at the exit screening.

Line 168: Of what is RMR the abbreviation? Please write it in full first with the    abbreviation in parentheses. Corrected Line 166

Line 170: What do you mean with the standard calculation? Which calculation is that? Corrected Line 181

Line 169, 171, 182, 211, 224, 231, 259, 264, 265, 270, 282, 295, 297: Please replace ‘our by ‘the’. Corrected throughout the entire paper

Line 173-174: During wWeight loss, tissue loss and metabolic 173 adaptations contribute to the  reduction in RMR. Corrected Line 178

Line 176-179: Was this statistically tested? Please add p-values.

Yes, the presented values were statistically tested for the total sample with comparisons between the first and third measurements (Table 3). Table 2 shows FATP and significant difference between group C - A p ≤ 0.029 and VFATL between groups A - B p ≤ 0.0000 and C - A p ≤ 0.0000.

Line 177-178: BMI values slightly diecreased in sub-groups B and C. but there was no change in group A group. Corrected Line 186

Line 183-185: Visceral fat is closely linked to overweight, obesity, and hypertersion, hyperlipidaemia, which can lead to serious chronic diseases, specifically diabetes type 2 and acute diseases such as myocardial infarction. Corrected Lines 192-194

Line 186-189: While adhering to a proper lifestyle in the form of an MD concerning weight reduction, the resulting bioelectric impedance measurement showed that cohorts B and C reduced their body weight by (-3.4 kg and / -2.1 kg) respectively. was this statistically tested? Please add p-values. Corrected Lines 196-198

Table 1.: Please add p-values.

Table 1. serves only as a descriptor of the overall sample.

Table 1: AGE (years) 

Corrected Line 203

Table 2: Are these values which report the differences correct? I assume this is based on the mean   values? Also, the p-values look really weird.

Yes, the values were based on average values. p-values are correct.

Table 2 and table 3: Are these correlation analyses or the results of the t-test? Because correlation analyses are not described in the statistical analyses section.

These are correlation analyses based on RStudio. We have added this information in the statistical analysis section.

Consistency in word use: Three-month Corrected Line 203

Consistency in numbers after the comma in table4. Corrected Line 215

Line 217: We can utter that MD it’s more of a guideline than. Corrected Line 228

Linz 221-223: By In contrast, there is a risk of excess calorie intake as a specific portion of foodstuffs which could lead to weight gain. [30] made a distinction between “Hunger and Appetites” author claim that we think and behave in a certain way.  Reformulate the second sentence in a correct sentence. Also give some examples of healthy but caloric foods. Modified Line 232

Line 227-229: Where perceive accuracy depends on interoception, which refers to the processes by which we sense, interpret, and integrate signals originating from within the body. There seems to be something wrong with this sentence. Modified Line 240

Line 233-234: Yet it remains unclear whether deficit or sensibility to interoception contributes to or is a consequence of weight gain of our the sample that needs further investigation. Corrected Line 247

Line 235: Maybe ‘an increase in’ should be a better statement than ‘a higher’.

Corrected Line 248

Line 237: … diabetes type 2, Corrected Line 250

Line 241: better for heavier subjects. [12]. Corrected Line 253

Line 242: …overweight people have not been…Corrected Line 254

Line 244: … of BMI of 27 kg.m-2 in the present our samples may. Corrected Line 262

Line 250: A study by [19] … Name the authors please instead of adding only the number of the  reference. Corrected Line 268

Line 256: …(MD diet), …Corrected Line 274

Line 258: Further [33] asserts… Name the authors please instead of adding only the number of the   reference. Corrected Line 281

Line 258: can increase the amount of how much visceral fat the our body. Corrected Lines 281-282

Line 261-264: Higher education attainment and socioeconomic status can be associated with better functional, healthy literacy, i.e., an individual’s capacity to obtain, process, and understand basic health information and services sufficiently to make appropriate health decisions [29]. This is not correct sentence. Make two sentences of it. Modified Line 287

Line 268: …related of having to getting insufficient sleep. Corrected Line 292

Line 276: (≥5% weight loss). Corrected Line 300

Line 284-286: Given that lifestyle modification (its domains) is challenging to initiate and maintain [14]. Weight loss possible, although difficult to conceptualize, is that weight loss it’s not entirely about the diet (eating less). These are not a correct sentence, please improve the language throughout the whole paper. Corrected Lines 308-313. Language was revised through the entire paper.

Line 291, 200: Our The present study. Corrected Line 316

Conclusion: the conclusion should be an overview of the results, while the recommendations for further research should be added to the discussion section. Revised Lines 324-327, 336-342

Line 331: Conflicts of Interest: The authors declare no conflict of intertest. Can be deleted because it is already mentioned at line 325-326. Corrected Line 336

Reviewer 2 Report

The paper consists of a nutritional intervention carried out on a population of overweight women.

In order to publish the work, the following considerations should be taken into account:

A revision of the English language is needed.

Even if the diets are personalized, the authors should indicate the main component foods.

Line 42: in the Mediterranean diet, red wine is consumed during the meal, not at the end of the meal

Lines 54-57: I consider that it is not appropriate to use the abbreviation of years (Y).

Line 76 indicate the meaning BMI.

The expression "certified nutritionists" is not correct.

Lines 94 and 95 the term probands is not correct.

Clearly state what a sparing diet consists of.

Lines 123 and 124 it is necessary to include commas to better understand the sentence.

Lines 138-140 Explain how a decrease in body weight increases IBM. Data do not match those in Table 1.

Line 140 "FATP did not change convincingly all along the experimentation in all cohorts, more specifically A group approximately 1%, B 2% respectively C 3% 1were higher than desirable for their age group category": Explain clearly what you mean by percentages.

Line 148 remove the brackets from Table 4.

Indicate in Table 1 whether the data are statistically significant.

Lines 233-234 according to the results in Table 1 there is no weight gain in any of the three study groups.

Author Response

Thank you for your valuable comment. We appreciate that. 

2 - Review

The paper consists of a nutritional intervention carried out on a population of overweight women.

To publish the work, the following considerations should be taken into account:

A revision of the English language is needed. Revised

Even if the diets are personalized, the authors should indicate the main component foods. Modified Line 115-118

Line 42: in the Mediterranean diet, red wine is consumed during the meal, not at the end of the meal. Corrected Line 55

Lines 54-57: I consider that it is not appropriate to use the abbreviation of years (Y). Corrected Line 74

Line 76 indicate the meaning BMI. Corrected Line 97

The expression "certified nutritionists" is not correct. Corrected Line 113

Lines 94 and 95 the term probands is not correct. Corrected throughtout the text

Lines 123 and 124 it is necessary to include commas to better understand the sentence. Corrected 164

Lines 138-140 Explain how a decrease in body weight increases IBM. Data do not match those in Table 1. Line 203

Body weight and BMI values are based on analysis of the BodyStatQuadScan 4000 TouchScreen. The values in Table 1 are correct. Only in group A a slight increase in body weight slightly reduce BMI, which can be explained by the fact that we reported average values. In the complete cohort (N=181) and groups B and C, the decrease in body weight and BMI values are evident.

Line 140 "FATP did not change convincingly all along the experimentation in all cohorts, more specifically A group approximately 1%, B 2% respectively C 3% 1were higher than desirable for their age group category": Explain clearly what you mean by percentages.

Line 156

The BodyStatQuadScan 4000 TouchScreen device reports the FATP parameter in %. Table 1 shows that, based on the personalized diet compared to baseline values, there was approximately a 1% reduction in Group A, a 2% reduction in Group B, and no apparent change in Group C. Based on the % FATP changes during the program, we wanted to highlight its effect.

Line 148 remove the brackets from Table 4. Corrected Line

Indicate in Table 1 whether the data are statistically significant.

Line 203

Table 1 serves as descriptors. Statistically, significant differences are shown in the following tables.

Lines 233-234 according to the results in Table 1 there is no weight gain in any of the three study groups. Line 203

There was an overall slight reduction in body weight in groups B and C; however, in group A, we observed a slight increase in body weight.

Round 2

Reviewer 1 Report

Line 17, 175, 177, 193, 197, 223, 336: try to avoid the use of the word ‘our’

Line 23-25: “Comprehensive analysis did not establish convincing evidence of the benefits of MD

on selected integrands” is written twice

Line 25-26: However, values of total body fat (FATP) between groups show a significant p ≤ 0,032 between groups A and C p ≤ 0.029, which can be attributed to the cohort’s age à try to put the p-values in parantheses and reform the sentence please

Line 27-28: Values in groups A and B p ≤ 0.000 and C and A p ≤ 0.000 indicated no changes in visceral fat (VFATL) à same suggestions as the line 25-26; p can’t be 0.000, please write ≤ 0.001 instead; p-values were significant though? But you say there were no changes. Is ‘changes’ the right word? Was this a correlation analysis? Please be more clear here.

Line 37: developing à development

Line 39:  …obesity-related consequences hypertension… à there seems to miss a word between ‘consequences’ and ‘hypertension’

Line 40-43: incorrect sentence

Line 55: …of the MD diets.

Line 64-65, 235, 237: try to avoid the word ‘we’

Line 75: following à due to

Line 77: x-x-years-old à with or without dashes?

Line 155-157: incorrect sentence

Line 158-160: same as line 155-157

Line 166-171: this does not belong in the results section, but in the discussion section

Line 172: if the FFM was stable, the TBW should also be stable, no?

Line 173-179: this does not belong in the results section, but in the discussion section

Line 185-186: idem

Line 191-197: idem

Line 234-236: incorrect sentence

Line 252: …disease, type 2, diabetes, …

Line 258: [23] à name the authors instead of a number

Line 263: (30-60-year-old)

Line 307: put reference [15] at the very back of the sentence

Author Response

Review 1

Line 17, 175, 177, 193, 197, 223, 336: try to avoid the use of the word ‘our’ - corrected

Line 23-25: “Comprehensive analysis did not establish convincing evidence of the benefits of MD on selected integrands” is written twice - corrected

Line 25-26: However, values of total body fat (FATP) between groups show a significant p ≤ 0,032 between groups A and C p ≤ 0.029, which can be attributed to the cohort’s age à try to put the p-values in parantheses and reform the sentence please - modified

However, total body fat (FATP) values between groups showed a significant difference (p ≤ 0.032) between groups A and C (p ≤ 0.029), which can be attributed to the age of the cohorts (30-39 vs. 50-60 years). Values in groups A and B (p ≤ 0.001) and C and A (p ≤ 0.001) are significant between groups, but do not indicated any changes in visceral fat (VFATL) in individual groups.

Line 27-28: Values in groups A and B p ≤ 0.000 and C and A p ≤ 0.000 indicated no changes in visceral fat (VFATL) à same suggestions as the line 25-26; p can’t be 0.000, please write ≤ 0.001 instead; p-values were significant though? But you say there were no changes. Is ‘changes’ the right word? Was this a correlation analysis? Please be more clear here – modified line 25 - 26

Line 37: developing à development – corrected

Line 39:  …obesity-related consequences hypertension… à there seems to miss a word between ‘consequences’ and ‘hypertension’

Some research [5, 6, 8, 30] has suggested that the Mediterranean diet (MD) acts as a hindering means in suppressing the consequences of hypertension, hyperglycemia, and dyslipidemia related to overweight and obesity [10, 25]. – corrected

Line 40-43: incorrect sentence

Esposito et al (2011) in his study Mediterranean Diet and Weight Loss: Meta-Analysis with 3,436 participants indicate significant effect of MD on body weight (mean difference between MD and control diet, - 1,75 kg, 95% CI; -2.86 to -0.64 kg) and BMI (mean difference, -0.57 kg/m2, -0.93 to – 0.21 kg/m2 ).corrected

Line 55: …of the MD diets. – corrected

Line 64-65, 235, 237: try to avoid the word ‘we’– corrected

Line 75: following à due to – corrected

Line 77: x-x-years-old à with or without dashes?

Yes with dashes because this reflects the age range of each group, for example, group A consisted of 59 women between 30 and 39 years of age.corrected

Line 155-157: incorrect sentence

All subjects (all three cohort groups) were overweight during the entire investigation. Body weight reduction was observed progressively in groups B and C [-3.4 kg (-4%) and -2.1 kg (-3%)]; body weight reduction was not detected in group A after the three-month follow up. BMI in the A group was 26,7 kg.m-2 with no improvement detected, in group B 27,9 kg.m-2 improvement 5% respectively C 27,8 kg.m-2 enhancement of BMI by 2,5% which was in congruence with weight losses of the sample.corrected

Line 158-160: same as line 155-157

All subjects (all three cohort groups) were overweight during the entire investigation. Body weight reduction was observed progressively in groups B and C (-3.4 kg-4% / -2.1 kg-3%); body weight reduction was not detected in group A after the three-month intervention. BMI in the A group was 26,7 kg.m-2 with no improvement detected, B 27,9 kg.m-2 improvement 5% respectively C 27,8 kg.m-2 enhancement of BMI by 2,5% which was in congruence with weight losses of the sample. FATP did not change convincingly all along the experimentation in all cohorts. However, we observed significant differences between groups A and C (p ≤ 0.032 vs. p ≤ 0.029).corrected

Line 166-171: this does not belong in the results section, but in the discussion section  - moved to line 311

Line 172: if the FFM was stable, the TBW should also be stable, no?

If a person is obese has a lower percentage of FFM and a higher percentage of FM it also translats to decreases in TBW.corrected

Line 173-179: this does not belong in the results section, but in the discussion section – corrected moved to Lines 865-871 (Word)

Line 185-186: idem  

Line 191-197: idem

An average female has a BMR of around 1410 kcal or 5,900 kJ. The BMR of our selection ranged from 1538-1497 (kcal). Weight loss, tissue loss, and metabolic adaptations reduce resting metabolic rate (RMR). BMR is usually slightly lower than RMR measurement. A more accurate Mifflin-St Jeor equation of RMR showed that the average BMR of the entire sample was 1323 kcal which is about 200 kcal lower than a calculation of BMR. Moreover, the average metabolic age (Harris-Benedict formula) of the entire sample was about five years higher than the actual age throughout the investigation (46 vs. 51), which indicates a need for lifestyle modification.

Assume this is what the reviewer was referring to. I do not see exact words repeated.

Line 234-236: incorrect sentence – corrected

Line 252: …disease, type 2, diabetes, …– corrected

Line 258: [23] à name the authors instead of a number …– corrected

Line 263: (30-60-year-old) …– corrected

Line 307: put reference [15] at the very back of the sentence …– corrected

Thank you for your valuable comment. We appreciate that. 

With respect 

I.U

Reviewer 2 Report

Dear authors, despite the proofreading work carried out by you, I consider that the work has many flaws in the wording, expression and interpretation of the results.

The following are some of the aspects that need to be improved: 

- Eliminate abbreviations from the abstract

- Throughout the text, when a result is indicated for a specific group, a colon should be included after the  letter (e.g.: A: 20 kg).

Lines 24-25: repeated in the text.

Lines 25-26: wrongly expressed statistical difference between groups

Line 43: cannot put an abbreviation for the first time without indicating its meaning (IQ, BMI)

Line 71: years-old, participated in this study over the three months, May-July 2022. 

Line 94: Wwere

Line 153 onwards: it is necessary to express the results correctly (-3.4 kg-4% / -2.1 kg-3%) (kg.m-2) 

Line 59: I don't understand this way of expressing significant differences, when there are differences between two groups only one "p value" appears and in this case they indicate two: "However, we observed significant differences between groups A and C. p ≤ 0.032 vs. p ≤ 0.029".

In my opinion, it is more interesting to make a statistical analysis within each group (A, B and C) of the evolution of the different parameters over the three months, rather than a correlation analysis between the groups; or a general analysis including all the study subjects.

Line 254 "obesity paradox." Delete the point.

Line 276: I don't understand this sentence: "resulted in an intentional behavior, i.e., MD."

280: This statement is repeated in line 167: Investigations [27,14] have shown that a low-carb diet is more effective at 280 reducing visceral fat than a low-fat diet. 

In the conclusion it should be clearly stated that the expected weight loss has not been observed.

Author Response

Review 2

Dear authors, despite the proofreading work carried out by you, I consider that the work has many flaws in the wording, expression and interpretation of the results.

The following are some of the aspects that need to be improved: 

Eliminate abbreviations from the abstract …– corrected

Throughout the text, when a result is indicated for a specific group, a colon should be included after the  letter (e.g.: A: 20 kg). - corrected

Lines 24-25: repeated in the text. – …– corrected

Lines 25-26: wrongly expressed statistical difference between groups …– corrected

Line 43: cannot put an abbreviation for the first time without indicating its meaning (IQ, BMI) – corrected

Line 71: years-old, participated in this study over the three months, May-July 2022. – corrected

Line 94: Wwere – corrected

Line 153 onwards: it is necessary to express the results correctly (-3.4 kg-4% / -2.1 kg-3%) (kg.m-2) – corrected

Line 59: I don't understand this way of expressing significant differences, when there are differences between two groups only one "p value" appears and in this case they indicate two: "However, we observed significant differences between groups A and C. p ≤ 0.032 vs. p ≤ 0.029".

The differences can be attributed to the age of the samples, not to the MD program.

In my opinion, it is more interesting to make a statistical analysis within each group (A, B and C) of the evolution of the different parameters over the three months, rather than a correlation analysis between the groups; or a general analysis including all the study subjects.

Based on the results (Table 1), we found no differences within each group. That is why we focus attention on comparison between the groups.

Line 254 "obesity paradox." Delete the point. corrected

Line 276: I don't understand this sentence: "resulted in an intentional behavior, i.e., MD."

Referring to the behavior that being intentional or purposive in our case voluntary participation in MD plan.

280: This statement is repeated in line 167: Investigations [27,14] have shown that a low-carb diet is more effective at 280 reducing visceral fat than a low-fat diet.  – corrected

In the conclusion it should be clearly stated that the expected weight loss has not been observed. – corrected

Thank you very much for your valuable comments and recommendations. We appreciate that. 

With respect

I.U